# Remittance Flows from Healthcare Workers in Toronto, Canada

Iffath Unissa Syed 

Department of Health Policy and Administration, Penn State University, Sharon, PA 16146, USA; ixs5254@psu.edu

**Abstract:** Previous research indicates that Canadian healthcare workers, particularly long-term care (LTC) workers, are frequently composed of immigrant and racialized/visible minorities (VM) who are often precariously employed, underpaid, and face significant work-related stress, violence, injuries, illness, and health inequities. Few studies, however, have analyzed the contributions and impact of their labor in international contexts and on global communities. For instance, it is estimated that over CAD 5 billion-worth of remittances originate from Canada, yet no studies to date have examined the contributions of these remittances from Canadian workers, especially from urbanized regions consisting of VM and immigrants who live and/or work in diverse and multicultural places like Toronto. The present study is the first to investigate health and LTC workers' roles and behaviors as related to remittances. The rationale for this study is to fill important knowledge gaps. Accordingly, this study asked: Do health/LTC workers in the site of study send remittances? If so, which workers send remittances, and who are the recipients of these remittances? What is the range of monetary value of annual remittances that each worker is able to send? What is the purpose of these remittances? What motivates the decision to send remittances? This mixed-methods study used a single-case design and relied on interviews and a survey. The results indicate that many LTC workers provided significant financial support to transnational families, up to CAD 15,000 annually, for a variety of reasons, including support for education and healthcare costs, or as gifts during cultural festivals. However, the inability to send remittances was also a source of distress for those who wanted to assist their families but were unable to do so. These findings raise important questions that could be directed for future research. For example, are there circumstances under which financial remittances are funded through loans or debt? What are the implications for the sustainability and impact of remittances, given the current COVID-19 pandemic and its economic effect of dampening incomes and wages, worsening migrants' health, wellbeing, and quality of life, as well as adversely affecting recipient economies and the quality of life of global communities?

**Keywords:** remittances; precarious work; racialized workers; feminist political economy; gender; race; immigrants; displacement and migration

## 1. Introduction

International financial remittances, estimated at USD 554 billion, to low- and middle-income countries reached a record high in 2019 [1]. Prior to that, it was USD 441 billion in 2015 [2] and USD 125 billion in 2004 [3]. The growth of these remittances in recent years has been explained by the migration of workers who send remittances, especially from wealthy nations [4–6]. The growth and development of remittances have generated significant interest and have prompted commercial banks to enter the market of monetary flows in order to profit from them [7].

For the purpose of this paper, remittances will refer to money that (im)migrants earn while working or living abroad, which they then send to recipients in their countries of origin or homelands, based on familial, cultural, or economic ties [8]. Remittances are important because of their relative size in recipient economies [9], and positive and significant outcomes for poverty alleviation [10] and financial development [11], which may help to sustain households and build health and wellbeing among socially and

materially disadvantaged communities around the world. Remittances are increasingly a significant source of external financing for developing countries, given that government-to-government foreign aid declined in the 1990s, and economic and financial crises have become more frequent and intense [7]. Research suggests that there are certain political effects of remittances, such as politicians actively campaigning abroad, as well as appealing to the global diasporas who are often the senders of remittances [7]. Countries receiving high levels of remittances have also been found to sustain higher fiscal deficits while pushing back on international financial organizations, such as the World Bank and the International Monetary Fund [1,12].

Remittances are considered to be one of the few reliable and sustainable sources of income for recipients [10] and are also becoming substitutes for foreign aid, given that they do not pose a burden on taxpayers [7]. Remittances are also important because large numbers of households may rely on them [7], and because they provide social and material protection to poor households [11], which reduces their vulnerability to economic shocks, crises, or other circumstances [7].

The recipients of remittances are often referred to as a transnational family. Transnational family members are those individuals who are left behind in their countries of origin [13]. Although transnational families are separated by international borders and geographies, as well as distances, they are held together through bonds of collective unity, welfare, and family-hood [14]. Transnational families can experience intra-state migration, such as the rural to urban movement within a specific country, or it can involve the migration of working-class people or professionals from South Asia, East Asia, Southeast Asia, as well as South America and the Caribbean, to North American and European cities [15–25].

While remittances amounting to CAD 5.2 billion originate from Canada [8], there is limited knowledge about remittances from Canadian workers, especially in sectors that are densely employed with those from immigrant and racialized backgrounds. Racialized people are visible minorities (VM) who experience the process of racialization, which is an active and ongoing process that results in unequal and unfair treatment of particular groups of women and men [26], and that has significant negative consequences on their lives and livelihoods [27].

There is currently limited knowledge about how racialized and immigrant workers are financially connected to their transnational families. The reason why this financial connection to transnational families is important is that it may reveal how remittances may serve to assist them. If there are indeed strong financial ties/connections to transnational families, then it also raises questions about the quantity of funds that are sent to them. Furthermore, the quantity or value could be indicators of the purpose of remittances. For example, sending CAD 500 occasionally may reflect gifts or meeting certain immediate or quotidian needs, such as food or medicine. However, larger amounts could be indicative of significant expenses, such as investments, renovations, tuition/education costs, or larger medical expenses, such as hospitalizations.

The purpose of this study is to investigate remittances from healthcare workers who are employed in an urban long-term care (LTC) home in Toronto, Canada. The motivation and rationale for selecting these workers are because they are often composed of immigrant or racialized populations [28,29]; the former may have retained connections and ties to their transnational families and may remit. Canada is a settler state, and a new home or host to many (im)migrants who have arrived from previous colonial states, the latter of whom were a part of labor diasporas [30]. Thus, it is reasonable to expect that some of them would send remittances internationally to their previous countries of origin and ancestral backgrounds. Indeed, research from the World Health Organization (WHO) suggests that immigrant healthcare workers do generally have the intention to send remittances back to their transnational families [31]. Remittances would be considered important to these workers because they can provide significant support to their global families and communities. For example, remittances impact and alleviate poverty in low- and middle-income countries, improve livelihoods and nutritional outcomes, solve food shortages,

improve spending on education, and reduce child labor in disadvantaged households [1]. Accordingly, the research questions for this study included the following questions: Do LTC workers send remittances? If so, which workers send remittances, and who are the recipients of these remittances? What is the range of monetary value of annual remittances that each care worker is able to send? What is the purpose of these remittances and what motivates the decision to send remittances?

*Theoretical/Conceptual Framework*

This study is informed by feminist materialist scholarship, which combines feminist theory with a political economy approach (as opposed to a libertarian one) [32], and this background guided the development of research questions, the analysis, and the interpretation of the findings. These lenses were selected because domestic work and healthcare work by older adults in LTC facilities and nursing homes tend to be performed primarily by women, immigrants, and racialized persons in the region of study [28,29,33]. Feminism is concerned with gender inequalities that arise from a system of patriarchy [34,35]. Feminists argue that society is gendered in such a way that women and men have fundamentally different experiences and access to power and privilege [34,35]. Feminists have both criticized and expanded upon materialist approaches in representing and considering women's perspectives [34,35]. Feminist materialist scholarship extends the ideas of materialism by connecting market relations with domestic ones [36]. Feminist political economy frameworks also focus on equity for women [37]. Scholars examine the economic needs of the family, the work of women in the home and in labor markets, and relationships within workplaces. Feminist materialist scholarship also examines tensions related to women's paid and unpaid work, such as how production and reproduction affect women's lives [38]. For instance, women's reproductive and unpaid caregiving roles could modulate the extent to which women participate in economic/paid economies, which can then affect their material conditions, social, political, health, and overall life circumstances. Therefore, it is important to consider the interactions between the micro, meso, and macro levels within which individuals, institutions, organizations and other structures are embedded [39]. Feminist perspectives have been influential in medical sociology, and in the sociology of women's health and illness, which is a supplemental approach [39].

A feminist political economy of health lens would reject the competing approaches of biological determinism and biomedical models because the latter assume that maleness and femaleness start, and usually end, with sex differences in reproductive systems that lead to differences in health and wellbeing [40–42]. However, the biological differences between women and men go beyond the obvious ones related to their reproductive systems and also include genetic, hormonal, metabolic, and other variations, including class, culture, and ethnicity [40–42]. Feminist political economy of health perspectives include the identification of multiple markers of difference, which provides a fuller picture of and context for the study participants [40–42]. These are often simultaneous experiences of gender, class, race, sexual orientation, size, or other social differences experienced by women [43]. Otherwise, the limitations of using single-identity markers, such as gender, immigrant status, or Aboriginal status, would lead to the false classification of people that does not reflect their lived realities [44]. Therefore, these perspectives are promising alternatives to a one-size-fits-all approach [34,35].

The feminist political economy of health was deemed to be appropriate for this study because it involved a largely female-driven workforce composed of racialized immigrants. Using this theoretical framework is helpful and relevant because of the demographic profile of participants since many care workers in urbanized regions in Canada are composed of people from rich ethnic, linguistic, racial, and immigrant backgrounds who may send remittances. Accordingly, examining multiple markers would illuminate the process of sending/receiving remittances. Immigrants are also rarely extracted from random populations, meaning that there are ethnic, religious, educational, regional, income, or other markers of difference among immigrants [7]. They migrate and settle in host countries

because of family ties or sponsorship, and/or because they possess certain desirable skills and educational qualifications. In Canada, immigration is determined by Citizenship and Immigration Canada, the government department responsible for immigration policy, and it targets certain immigrants due to their work experience and potential economic contributions to Canadian society [45]. Official data suggest that 21.9% of Canada's population is composed of immigrants [46]. In Toronto, which is the region where this study was undertaken, immigrants represent 46.1% of Toronto's population [47].

A feminist political economy of health lens [33] can be combined with anti-racist and other critical research approaches to help to inform a growing body of research about racialized immigrants, and the processes of resistance to, for example, cultural assimilation [48,49] that conforms to hegemonic norms and values [50]. Supplementary approaches can also be considered with respect to the remittance literature, including work/labor studies and the sociology of migration. Extending these approaches in order to analyze class, gender, and race may help explain some impacts of racialization on marginalized men's and women's lives. In other words, they would help to demonstrate the complex nature of human interactions, experiences, and consequences. Accordingly, these frameworks and approaches guided the collection of data, analysis, and the presentation of findings, and were implemented in this study in several ways. Firstly, they were implemented by asking questions about background, birth status, sex/gender, visible minority VM status, or ancestral background. Secondly, they were implemented by guiding the analysis with the use of an appropriate coding system, and thirdly, by stratifying the findings by sex, VM status, immigrant status, and so forth. Finally, the researcher attempted to be continually reflexive during all stages of the research process, and reflected on ways in which the researcher's background, gender, culture, history and socioeconomic origin informed the data collection and analysis procedures.

## 2. Materials and Methods

This mixed-methods study is derived from a larger research project that had a single-case study design and relied on qualitative interviews, observations, and a survey. Because this case study used a combination of these sources of evidence in a single study, it is considered to be a mixed-methods approach [51,52] p. 17, which has a higher rating for quality than those case studies that rely "[ . . . ] on only single sources of information" [52], p. 119.

### 2.1. Data Collection

The primary data collection technique for this study was via qualitative, in-depth, key informant interviews. Face-to-face, in-depth, semi-structured interviews were conducted with 42 participants, digitally recorded, and transcribed. The interviews were carried out in private office spaces away from the main units/wings at the site of study, which enabled good rapport and trust with participants. Participants were asked for clarification on any points that were unclear. For the quantitative component, an exploratory, paper-based pilot survey was distributed to participants concurrently during site visits, as well as a separate demographic questionnaire following each interview. The survey included questions about remittances with an upper limit set as CAD 500, since remittances over CAD 500 were not expected to be sent by most LTC workers, given that it is known in the literature that many workers have modest incomes. In total, 92 respondents filled and returned the survey from a pool of 176 participants, i.e., a response rate of 52% was achieved. One survey response was excluded from the analysis because the worker was not responsible for work that was related to any aspect of the LTC home. Participants were provided with incentives in the form of a gift card for their participation in the interview, and a chance to enter a raffle for a larger gift card for completing the survey. The incentives were small enough that they did not provide the sole motivation for individuals to participate in the study; rather, participants would have a desire to participate regardless. Thus, the incentives should not

have had a significant effect on participation or data quality. The research was approved by the institutional Office of Research Ethics (ORE) (ethics approval number: STU2016-139).

*2.2. Measures*

For the qualitative component, multiple units of measure were organized according to workers' sex, job titles or roles, visible minority (VM) status, full-time (FT) status and part-time (PT) work status, among other things. VM status was derived from participants' responses to certain interview prompts, such as "Tell me about your background". Participants' characteristics are listed in Table 1.

**Table 1.** Interview participants' characteristics (*n* = 42) Source: [53].

| Characteristic | | Frequency | % |
|---|---|---|---|
| Sex | Female | 35 | 83.3% |
| | Male | 7 | 16.7% |
| Employment Type | Full-Time | 32 | 76.2% |
| | Part-Time | 10 | 23.8% |
| Race/VM Status | Non-VM, Non-racialized, White | 12 | 28.6% |
| | VM, Racialized | 30 | 71.4% |
| Job Title/Role | Trainee | 3 | 7.1% |
| | Allied Health | 7 | 16.7% |
| | Nurse | 9 | 21.4% |
| | Manager | 4 | 9.5% |
| | Support Staff | 6 | 14.3% |
| | Ancillary | 6 | 14.3% |
| | PSW | 7 | 16.7% |

VM status is an important marker of difference and may be a proxy or measure of migration among particular ethnic groups/ancestral backgrounds. It is expected that remittance behaviors may be different depending on VM status. In other words, it is anticipated that remittances may be reported more frequently among VM than non-VM participants. For the survey, the purpose was to examine the effect of birth/immigration status on funds sent to family or relatives abroad. The survey asked the respondents about the quantum of remittances sent (if applicable), and also collected additional data, such as birth/immigration status, which was used for a Mann–Whitney-U test. Groups were categorized as born in Canada, born outside Canada, and allocated as the independent variables, and international remittances sent abroad were allocated as the dependent (scale) variable.

*2.3. Analysis*

Although qualitative and quantitative data collection were carried out concurrently, there was "separate data analysis" [54] that occurred individually, and each component is presented in the next section separately, as is consistent with the analysis. The quantitative component was intended to report descriptive statistics, among other things, and to help bring clarity to the findings, while the qualitative component augmented and unpacked some of the experiences. For the qualitative data analysis, field notes and interview transcripts were analyzed with thematic analysis for the study, using a coding system with the aid of the NVivo computer software program to organize and sort information. These themes are organized by subheadings in the results section. Quantitative statistical data analysis of the demographic questionnaire and survey occurred with the assistance of Excel and a quantitative statistical software program (IBM SPSS Statistics, 26.0, Armonk, NY, USA).

## 3. Results

### 3.1. Characteristics of Workers Who Send Remittances

Approximately 48.9% of the survey respondents ($n = 43$) indicated that they sent their income internationally, while the remainder indicated that this scenario was not applicable to them (Table 2).

**Table 2.** Reporting of annual international remittances, stratified by immigrant status. Almost half (48.9%, $n = 43$) of the survey respondents sent their income internationally (i.e., remittances to family), while 51.1% ($n = 45$) of the respondents did not do it.

| | Total | | Non-Immigrant | | Immigrant | |
|---|---|---|---|---|---|---|
| **Remittances Status** | **Frequency** | **%** | **Frequency** | **%** | **Frequency** | **%** |
| Not Applicable/Not Sent * | 45 | 51.1% | 18 | 40.0% | 23 | 51.1% |
| Applicable * | 43 | 48.9% | 1 | 2.3% | 41 | 95.3% |
| Total * | 88 | 100.0% | 19 | 21.6% | 64 | 72.7% |

\* A total of 5 respondents did not disclose their immigrant status.

Of those who did send remittances ($n = 43$), 76.7% ($n = 33$) sent over CAD 500 annually, and of these individuals who sent over CAD 500, 97% ($n = 32$) were immigrant, 87.9% ($n = 29$) were female, and 90.9% ($n = 30$) were VM (Tables 3–5).

**Table 3.** Amount of annual international remittances, stratified by immigrant status. Of those who did send international remittances ($n = 43$), 76.7% ($n = 33$) sent over CAD 500 annually. The remainder ($n = 10$) sent between CAD 0 and CAD 500 (23.3%). The survey did not ask the purpose of these remittances.

| | Total | | Non-Immigrant | | Immigrant | |
|---|---|---|---|---|---|---|
| **Remittance Range/Amount** | **Frequency** | **%** | **Frequency** | **%** | **Frequency** | **%** |
| Less than CAD 500 | 10 | 23.3% | 1 | 10.0% | 9 | 90.0% |
| Over CAD 500 * | 33 | 76.7% | 0 | 0.0% | 32 | 97.0% |
| Total * | 43 | 100.0% | 1 | 2.3% | 41 | 95.3% |

\* 1 respondent who reported remittances did not disclose immigrant status.

**Table 4.** Amount of annual international remittances, stratified by sex. Of those who sent remittances ($n = 43$), 86.0% ($n = 37$) were female.

| | Total | | Female | | Male | |
|---|---|---|---|---|---|---|
| **Remittance Range/Amount** | **Frequency** | **%** | **Frequency** | **%** | **Frequency** | **%** |
| Less than CAD 500 | 10 | 23.3% | 8 | 80.0% | 2 | 20.0% |
| Over CAD 500 * | 33 | 76.7% | 29 | 87.9% | 3 | 9.1% |
| Total * | 43 | 100.0% | 37 | 86.0% | 5 | 11.6% |

\* 1 respondent who reported remittances did not disclose sex.

**Table 5.** Amount of annual international remittances, stratified by VM status. Of those who sent remittances ($n = 43$), 81.4% ($n = 35$) were VM.

| | Total | | Non-VM | | VM | |
|---|---|---|---|---|---|---|
| **Remittance Range/Amount** | **Frequency** | **%** | **Frequency** | **%** | **Frequency** | **%** |
| Less than CAD 500 | 10 | 23.3% | 0 | 0.0% | 10 | 100.0% |
| Over CAD 500 | 33 | 76.7% | 3 | 9.1% | 30 | 90.9% |
| Total | 43 | 100.0% | 8 | 18.6% | 35 | 81.4% |

Remittances were also analyzed according to the respondent's ancestral background, and senders were as follows: 25.6% ($n = 11$) were South Asian, 25.6% ($n = 11$) were Caribbean, 14.0% ($n = 6$) were African, 11.6% ($n = 5$) were Southeast Asian, 9.3% ($n = 4$) were East Asian, 7.0% ($n = 3$) were European, and so forth (Table 6).

The results of the Mann–Whitney-U Test indicated that participants born outside of Canada (i.e., the immigrant group) sent international annual remittances more frequently compared to those participants born in Canada ($U = 985$, $p < 0.001$) (Figure 1). This result is important because people born outside of Canada may not always have strong ties to their transnational families. For instance, it is expected that if some individuals immigrated as children with their immediate family, and were raised in Canada, they may have less robust ties to their countries of birth. However, these findings show that most immigrant workers have strong financial ties and send remittances.

**Independent samples: Mann–Whitney U test by Birth Status**

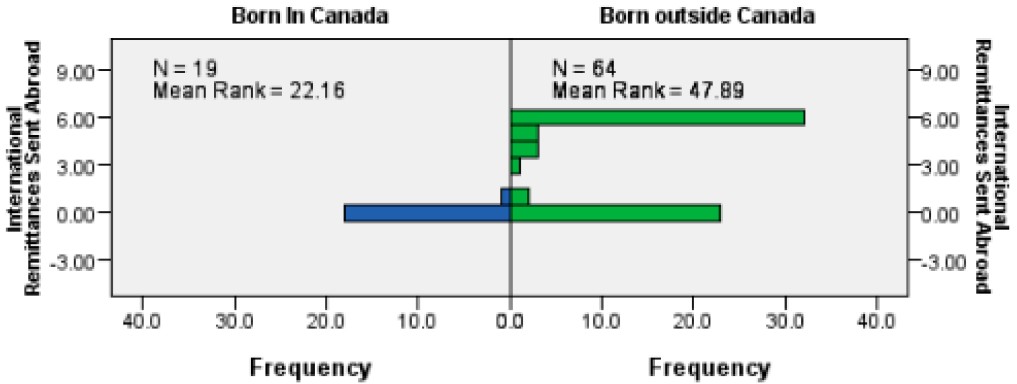

**Figure 1.** Mann–Whitney U test of international remittances by birth/immigration status.

A Mann–Whitney-U test was conducted using birth/immigration status. Groups were categorized as born in Canada, born outside Canada, and allocated as the independent variables, and international remittances sent abroad were allocated as the dependent (scale) variable. The purpose was to examine the effect of birth/immigration status on funds sent to family or relatives abroad. The results indicated that participants born outside of Canada (i.e., the immigrant group) sent international remittances more frequently compared to those participants born in Canada ($U = 985$, $p < 0.001$).

To examine the effect of gender, Mann–Whitney U tests were conducted to assess the effect of immigration status and racialization separately for male participants and female participants. The analysis indicated that female participants born outside of Canada sent income internationally more frequently than those female participants born in Canada ($U = 559.5$, $p < 0.047$) (Figure 2). The analysis was also conducted to test the effect of race/ancestral background; however, there were no significant differences in the frequencies of international annual remittances (data not shown). This result is important because it is often thought that the senders of remittances are male. For instance, in certain cultures, males are considered the main breadwinners of their families; thus, it is anticipated that men would send remittances more frequently than women. However, these findings show that women have strong financial ties to their countries of birth.

**Table 6.** Amount of annual international remittances, stratified by ancestral background. Remittances sent, according to Tables 2–5. 6% (*n* = 11) were South Asian, 25.6% (*n* = 11) were Caribbean, 14.0% (*n* = 6) were African, 11.6% (*n* = 5) were Southeast Asian, 9.3% (*n* = 4) were East Asian, 7.0% (*n* = 3) were European, etc.

| | European | | South Asian | | East Asian | | Southeast Asian | | Central Asian | | Caribbean | | African | | Other, e.g., South American | |
|---|---|---|---|---|---|---|---|---|---|---|---|---|---|---|---|---|
| **Remittance Range/Amount:** | **Freq** | **%** | **Freq** | **%** | **Freq** | **%** | **Freq** | **%** | **Freq** | **%** | **Freq** | **%** | **Freq** | **%** | **Freq** | **%** |
| Less than CAD 500 | 0 | 0.0% | 2 | 6.1% | 2 | 6.1% | 2 | 6.1% | 0 | 0.0% | 2 | 6.1% | 1 | 3.0% | 1 | 3.0% |
| Over CAD 500 | 3 | 9.1% | 9 | 27.3% | 2 | 6.1% | 3 | 9.1% | 1 | 3.0% | 9 | 27.3% | 5 | 15.2% | 1 | 3.0% |
| Total | 3 | 7.0% | 11 | 25.6% | 4 | 9.3% | 5 | 11.6% | 1 | 2.3% | 11 | 25.6% | 6 | 14.0% | 2 | 4.7% |

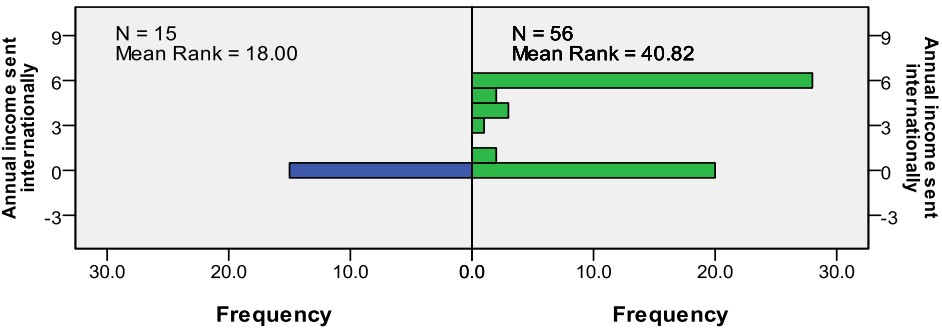

**Figure 2.** Mann–Whitney-U test of international remittances by birth/immigration status and sex.

A Mann–Whitney-U test was conducted to test the effect of sex/gender and birth/immigrant status on funds sent to family or relatives abroad. The results indicated that female participants born outside of Canada (i.e., the female immigrant group) sent international remittances more frequently compared to female participants born in Canada ($U = 559.5$, $p < 0.047$).

The interview data revealed interesting insights about the recipients of remittances, the purpose of remittances, the desire and ability to send remittances, and the moral/ethical meanings tied to the process of sending remittances.

### 3.2. Qualitative Findings

The analysis of the interview data resulted in five major themes that emerged as follows: common recipients of remittances; typical purpose of remittances; frequency of remittances; budget, desire, and ability to send remittances; and moral or ethical meanings connected to remittances.

### 3.2.1. Common Recipients of Remittances

The interview data suggest that the recipients of remittances included spouses, children, and other family or relatives, such as siblings, parents, and in-laws. These individuals are the transnational families of the participants, and many of them live in the participants' countries of origin. Common themes about recipients were descriptions that were tied to a particular need, e.g., older adults and aging parents who require assistance with healthcare or medical costs (versus supporting siblings, young children, or others).

### 3.2.2. Typical Purpose of Remittances

The purpose of the remittances varied among participants, depending on for whom the remittances were intended. Common reasons why workers sent remittances to their transnational families included support for the costs of living, specific medical or healthcare expenses, education-related costs, cultural or religious festivals, income supplementation, or non-specific amounts, such as gifts. Remittances were not allocated for religious purposes, or for the support of villages, clans, etc. However, a few workers did state that remittances were sent as investments.

Consider the situation below in which a racialized ancillary healthcare worker reported that he sent substantial sums overseas to Asia each year because of his mother's health issues and also for special occasions, such as festivals:

"Yearly, I think $13,000/$12,000 has to be sent for a festival [ . . . ] and recently my mom is sick, I gave them some money. They don't need, [ . . . ] I have to do this and

you know, the festival, we have a lot of relatives, my side and my wife's side, so this is our culture, before the festival and before they eat we send everybody some money. So $15,000/$12,000 every year has to go." (Participant 42, ancillary worker, male, VM, F/T).

In another example, a racialized manager said her husband's relatives (i.e., in-laws) who live overseas are supported by her husband, and her parents are supported by her. She indicated that their countries of habitation lack social and health services, such as the provision of medical care. Essential services in those countries are paid privately, and out of pocket:

"My husband's family, because he's basically the only one that supports, sends them money, and I'm the support for my parents. I have brothers that support them but we all kind of pitch in and help to make sure that their payments are because my dad is retired. My mom never worked. He was a bank manager but back home there's no health support, like you have to pay for going to hospital, doctor visits, medications, all of that, and standards of living there is so expensive even though it's a tiny country". (Participant 32, manager, female, VM, F/T)

A few participants also stated that they sent remittances as an investment paid in advance because they wanted to return to their country of origin to live there:

"I intend to go back there to live, so I want to pay forward also, 'cause when I get there who knows what life will dish out for you". (Participant 31, PSW, female, VM, F/T)

### 3.2.3. Frequency of Remittances

Another common theme was that many workers frequently and regularly sent remittances, although a few also said they did so only when requested. For the former group of workers, these findings could mean that their families relied on remittances for particular things. For example, participants 29, 31, 32, 40, and 42 reported that they sent regular money, and the support was for basic necessities such as food, shelter, and education. Participants 2, 35, and 42, however, suggested that remittances were temporary, intended as a gift, and were not regular obligations but were rather based on temporary need and cultural events or festivities. This means that transnational families experienced various levels of socioeconomic deprivation and need, which would depend on political, cultural, geographic, or other factors, such as the extent of social/welfare programs and policies of recipient countries. Recall that participant 32 indicated that she supported her transnational family in her homeland regularly to assist with out-of-pocket healthcare expenses, such as physician's visits, medications, and hospital stays. An ancillary worker (Participant 40, ancillary worker, female, VM, P/T) also sent money to her mother and sisters overseas on a monthly (regular) basis. However, after ending a relationship and being reduced to one income, she was only able to contribute on "special occasions". The latter meant that regular remittances were contingent upon sufficient levels of income. Another worker (Participant 2, allied health worker, female, VM, F/T) said she occasionally sent remittances to relatives overseas as gifts, but she did not consider this to be a regular obligation or a burden because all of her siblings were located outside of the country of origin, and everyone shared costs in the support provided to these overseas relatives.

Consider a further example below in which a racialized allied health worker said she sent money overseas to Asia when family members experienced an exceptional need:

"I have family back in [overseas country], but . . . I support them, but not on a regular basis. Yeah, not every month; but when a need arises, then I support them". (Participant 35, allied health worker, female, VM, F/T)

### 3.2.4. Budgets

The data also revealed a number of common themes about budgets. For the most part, remittances significantly affected senders' budgets, and senders admitted that they had to make certain sacrifices. This is an important finding because it is well known that care workers are underpaid, yet many exercise generosity despite the fact that they receive little in return. Consider the situation below in which a racialized manager was asked about

budgets. The surprising response was the fact that remittances were one of the biggest line items. This worker indicated that both herself and her husband felt a "stretch", financially, and that they were stressed about it:

I: "Okay. Tell me about what consumes the biggest part of your budget then?"

A: "Sending money for family back home [ . . . ] We have lots of families to support plus like the immediate family members plus extended family members that we have to help. So we always have to send money and that takes up a portion plus the payment here like the bills, the mortgage, I have one now in university, as second one going in this year actually, 2017 September. Mamma mia. So we kind of felt the stretch this year or last year, 2016 and we're kind of going to feel it more this year, so we're a bit on the stressed side when it comes to payment, but you know, we're thankful we're healthy. We can manage". (Participant 32, manager, female, VM, F/T)

In another example, a participant reported that she sent money to support both her family and also her husband's family, in which one member experienced a health crisis and illness. She suggested that the size of the remittances were so large, that an entire salary was exhausted, meaning that it was completely dedicated for this purpose:

"I help my husband's family in [overseas country], which we constantly supported them and sent them money and unfortunately sent them more and more because his brother became sick—cancer, last stage. So it means—it will cost us a lot of money and what I can say [ . . . ] From first day I helped my mom. I need to send money to my mom, to my brother's family, and my husband's family. So we were supporting two families. So one salary was going out". (Participant 29, manager, female, non-VM, F/T)

A racialized allied healthcare worker shared her experience in which there was no pre-determined budget but, rather, remittances were tailored according to need:

"It's never like a pre-planned budget because whenever there's a need, then we provide. So we kind of adjust and cut down on our expenses, yeah". (Participant 35, allied health worker, female, VM, F/T)

### 3.2.5. Desire and Ability to Send Remittances

Another common theme was that participants often expressed emotions about sending remittances, such as individuals who had the desire to send remittances and were able to do so (versus those who wished to send money but were unable to do so). This discrepancy largely depended on the socioeconomic status (SES) of senders (i.e., job title, full-time or part-time status, which would correspond with higher or lower salaries/incomes); with higher SES groups likely having the ability to support larger expenses. Being unable to send remittances, however, could be a source of discomfort and stress for participants and contribute to feelings of helplessness.

Consider the example below, where a racialized ancillary worker indicated that he had a sister in financial need overseas, but he was unable to help her:

"My sister in the [overseas country] [ . . . ] sometimes she is calling me and she needs the money, and I cannot afford to give—send her money [ . . . ] Even a penny, I cannot send [ . . . ] It's very tough. It's very hard". (Participant 12, ancillary worker, male, VM, F/T)

### 3.2.6. Moral and Ethical Meanings Connected to Remittances

Interview data also revealed a common theme about the moral obligations of sending remittances to help the less fortunate, and for poverty alleviation. This is an important finding because care work and care workers, the former of which is known to be gendered and poorly remunerated, are at odds. Care workers are poorly remunerated when they exchange their labor power for wages, yet they exercise generosity, despite receiving little in return. Why do care workers (and racialized/immigrant women and men) do this? The data suggest that, for some care workers, remittances had moral and ethical meanings that were tied to belief systems. For example, some participants mentioned belief systems that motivated them to send remittances, despite the fact that they experienced difficulties in assisting their families. However, they did so anyway because they felt it was rewarding

and it was their moral responsibility. Consider the racialized female personal support worker (PSW) below, who described remittances as a moral obligation to help others who are less fortunate, and also mentions her belief that she will be blessed for this endeavor:

"I send money home to my family. [ . . . ] because of economic hardship, I used to support my grandmother [ . . . ]. After my grandma pass on, I still continue to help them [the rest of the family] because they need the help [ . . . ]. So maybe my cousins' children might remember I used to help them because I pay for them to go to school, I help them with food, I help them with housing wherever I can. [ . . . ]But I - we grew up always helping each other, so it's like an obligation that you have to help the other ones that are not fortunate. You're fortunate to come out, so you share what you have. We always believe that you get blessed even more by sharing the little that you have". (Participant 31, PSW, female, VM, F/T)

Moral arguments about sending remittances raise an interesting and important point, given the gendered and racialized nature of care work. Previous studies emphasizing feminist perspectives suggest that care work is underpaid and undervalued because it is considered to be women's work, and that good women care for their families and others, either uncompensated or poorly paid, and in doing so, they attain feminine moral worth [55]. Women's work in the home or in the labor market is morally and ethically elevated in this regard, but it can be detrimental to women's material conditions because it is often invisible [56] or undervalued [29]. Data from this study suggest that remittances are no exception, and that women who carry out care work are expected not only to care for and share with their immediate family, but also geographically distant relatives and the transnational family that is afar. It is also important to mention that there may also be some aspects of reverse causality. For example, (im)migrant and racialized workers leave their home country for a variety of reasons, and there are a number of push-and-pull factors to consider, including higher remuneration, better working conditions, a better quality of life, and so forth. It is plausible that assisting transnational families is also a push factor, and being near their transnational families could make the situation worse than the current one in which there is assistance provided to the transnational family.

Another important point is that previous research shows that the senders of remittances are often male and are the most significant contributors [25]; however, the data from this study show that some of the most important contributions to households in the form of remittances and financial support to families for costs related to shelter, education, and health care, targeting places outside of Canada, originated from both female and male immigrant/racialized labor in the care work sector. The employment of female workers was historically considered to be supplemental income under a traditional male-breadwinner model that considered male workers to be the main income earners. However, this notion that women's work is supplemental and, therefore, negligible is challenged by the findings from this study, which show significant levels of support and financial assistance to the transnational family that originate from female immigrant and racialized workers. These findings, nevertheless, warrant further investigation, given that this sector of employment is potentially populated by female minorities, which may be driven by this context.

## 4. Discussion

The aim of this study was to investigate remittance behaviors among healthcare workers who are employed in an urbanized LTC home in Toronto, Canada. The study asked if LTC workers sent remittances, and indeed the findings suggest that 48.9% of workers did this. Another set of objectives was to analyze which workers sent remittances, and delineate who are the recipients of these remittances. The findings suggested that 95.3% of workers who sent remittances were immigrants, 86.0% were female, and 81.4% were VM, and that remittances were sent to older adults and aging parents, in-laws, siblings, and young children. Another objective was to gain knowledge of the monetary value of annual remittances, which was often well over CAD 500 but also depended on the purpose and

motivation to remit, such as for medical/healthcare costs, educational costs, investments, gifts to the transnational family for festivals, and so forth.

The key findings from this study confirm previous work that remittances originated from migrants [2,9]. This study also suggests that these migrants were often racialized, with most originating from particular regions, such as Africa, the Caribbean, and South Asia. Given that the majority of remittances were sent by female immigrants who were predominantly VM from Asian countries, this study also supports previous findings that remittances have a north-to-south flow. Remittances were sent to recipients and transnational families for a variety of reasons. The research from this study supports previous literature that, indeed, remittances fulfilled some of the transnational family's needs of daily consumption, such as food, shelter, clothing, education, and healthcare [8].

Taken together, the research from this study suggests that the value of remittances varied, and frequently exceeded CAD 500 annually, which was set as an initial threshold in the surveys. This finding was surprising, as this value was initially expected to be an upper limit, and remittances over CAD 500 were not expected to be sent by LTC workers, given that they are underpaid [57–59], lack basic paid sick days [60], and are often precariously employed [29], having to make difficult choices for basic necessities, such as selecting healthy food over nutritiously deficient food [61]. However, this study showed that annual remittances were as high as CAD 15,000, depending on the SES of senders using a proxy of job title and full-time or part-time work.

The fact that LTC workers are often underpaid and undervalued seems to contradict the significant contributions that LTC workers make, not only to the lives of elderly residents/recipients of care, but also to their transnational families and communities in the international context. In other words, many LTC workers are providing significant social and economic support to recipient economies, which is ironically rendered as invisible as their caregiving work. Accordingly, this study establishes a starting point for a deeper examination of these issues for future research in this sector. For example, this study showed that an annual amount over CAD 500 was the most frequently sent quantity of remittances; however, in some surprising instances, it reached 30 times this value at CAD 15,000 per annum. While this study does not investigate whether or not these remittances may be too high, it raises some rhetorical questions: is it fair that precariously employed, low-wage workers often bear the responsibility of supporting transnational families? Previous research shows that immigrants and racialized groups are often working in precarious [29], low-waged [27], dangerous [62], stressful, and unhealthy conditions [45,63,64] that often result in chronic illness and ill health [45]. Furthermore, this work raises moral and ethical implications regarding neoliberal state policies, which have cut back on foreign aid, resulting in the onus of assistance to those living in states with significant financial deficits falling on the shoulders of care workers. It seems that workers have few alternatives when there is an obligation to remit. Given that care work pays little, and requires significant work, it essentially means that care workers are giving more to others than they actually receive in return. Accordingly, workers may be sacrificing their own comfort, leisure, wealth, health, and wellbeing for the sake of their transnational families.

The theoretical explanations of the findings need to be interpreted with caution since it is a cross-sectional design. There were also some limitations to this study. For example, due to the exploratory design of the survey questionnaire, participants were not asked about the frequency of remittances in a given year, which could otherwise shed more light on the type of need and the ability of senders. For instance, a high frequency could mean high recipient need (and better ability/capacity of senders to remit); whereas a low frequency could mean low recipient need, e.g., occasional gifts (or lower ability/capacity of senders to remit). The qualitative data did reveal some patterns about frequencies in sending remittances, and that the reason why some types of remittances were frequent/regular was indeed due to high need, whereas irregular/low frequency was due to low need. Another limitation was that participants in this study who revealed that they were immigrants were not asked about their length of stay, which could otherwise shed more light on their

decisions to send remittances. Previous research suggests that the length of stay abroad influenced decisions to remit; specifically, the longer migrants stay abroad (or have a longer length of stay in a host country), the more likely they will remit to recipients in their home country, although the incidence decreases eventually for long absences/long lengths of stay [9].

It is also understood that remittances could finance the following: school attendance, especially for girls [65], as well as the purchase of land/buildings, fund wells and irrigation works, and provide liquidity for small enterprises in countries where there is a lack of functional credit markets [7]. However, this study did not investigate these areas.

Another limitation is that this study did not fully capture all of the possible relationships between the senders and recipients of remittances. For instance, mothers who leave their child(ren) in order to do caregiving work abroad may be more likely to remit large, frequent sums of money for their children [7]. These types of relationships were not captured in this study. Furthermore, remittances also affect communities that have a high dependency on them, and household members could simply stop working and wait for remittances on a monthly basis, which could suggest preferences to remain unemployed as members wait for the possibility to migrate themselves, rather than taking up jobs in the local market [7]. Research indicates that forced remittances have also been previously documented [66]. This study did not explicitly ask if remittances were forced, as it was beyond the scope of this study. However, the workers' narratives about the remittances seemed to suggest that forced remittances were rare. Finally, the sample sizes for the statistical analysis were small, and for the interview data, there were limited unique responses about remittances. For example, for some participants, there was no new or additional unique information. For others, remitting did not apply for them. Additionally, sometimes interviews were cut short because workers had to start their shifts or resume their work if they were on breaks, or due to sudden disruptions, e.g., on-call duty or other care requirements for residents, and they were not available subsequently for additional interviews. Another limitation was that the data and findings from both qualitative interviews and the survey questionnaire need to be interpreted with caution, since this sector of work largely employs female workers and offers easier accessibility for non-native female workers than foreign male immigrants, and there was a very limited sample size of male workers. Therefore, this study is only the first step of an introductory analysis on the issue of remittances.

This study was also conducted as a single case, which means that it is not always representative/generalizable to broader environments. However, one strength of focusing on one site means that certain issues can be investigated in deeper ways through a closer examination of context. The findings from this study provide some insight and a good starting point for in-depth accounts of the senders' perspectives. Furthermore, this study contributes new research that evolves from LTC work. To date, no study has examined the financial remittances of LTC workers.

*Migrant Health and Quality of Life*

Previous Canadian research of female immigrants and racialized workers suggest that these groups are often predisposed to chronic illness [29,45,50], and difficulties in managing domestic obligations [33] due to the precariousness of their work [29], which adversely affects their health and quality of life [45]. In addition, research has shown that in the care-work sector, many of these employees experience work hierarchies and strict divisions of labor [28], while regulating emotional health and wellbeing through various mechanisms [61]. This study confirms previous research about the health and wellbeing of LTC workers, and reveals new information about financial circumstances, which alludes to the quality of life not only of their transnational families but also their own—how they are managing and juggling multiple commitments with challenging work experiences. These findings shed light on under-researched areas of how financial and other processes are navigated by workers so as to engage with their families and provide socio-economic support. Understanding these practices may point to possible directions in shifting the

current framing of migrant health discourses that otherwise may not be reflective of ethnic, racial, or cultural groups and the complexity of their circumstances.

Based on the foregoing, investigating and discussing the value of remittances in deeper ways may be a starting point for further work. For future research, CAD 500 might be used as a baseline or minimum threshold, and quantities that are higher than this amount could be investigated in deeper ways. For example, remittances are used to buy land, homes, investment in businesses, or set aside as savings [8]. Given that many participants reported insufficient incomes, one could also investigate how remittances are funded. For instance, are remittances funded through debt? Indeed, this particular finding has been found in previous research [25].

Another area for future research might be examining the relationships between remittances and how they are shaped by social relations and cultural values, which was also previously investigated to some extent [25]. Furthermore, one could also examine how remittances are sent. For example, are they sent through networks of trusted agents? Who are these agents? Research suggests that the use of informal and sometimes underground channels persist, such as the hawala and hundi in South Asian countries that go back centuries, despite the increase in formal transfer mechanisms [7].

Another question to consider is: when does the process of sending remittances stop? For example, does it stop when family re-unification occurs, or does it stop when there is a death of the recipient, which have both been reported previously [25]? There are also gendered perspectives that can be investigated, such as the perception that money is a medium to care for families and sustain relationships in the home country, as well as gendered patterns among senders or recipients. For instance, previous research suggests that senders are often male [25]. Finally, given the significant economic impact of the COVID-19 pandemic on both host and recipient economies, the fluctuation or effect on sending/receiving remittances would need to be explored.

## 5. Conclusions

A number of key findings have arisen from this work. This is the first study to report on international financial support stemming from Canadian LTC, and it makes an original contribution to the fields of work/labor studies, feminist theory, and expands upon the remittance literature. The findings from this study contribute new information about economic/financial support remitted by LTC workers in Canada. These healthcare workers assisted families and relatives, and sent international remittances, which were more likely to be reported by women, VM, and workers born outside of Canada (i.e., immigrants) rather than men, non-VM, and Canadian-born workers.

Secondly, these findings not only contribute to the feminist political economy of health literature about female immigrants' global economic activities that are generated between their current places of residence and their countries of birth [67], but also the literature about transnational families. There is some literature about migrant workers' contributions to elderly care, and the fact that racialized immigrant women tend to do the work in the lowest sectors of the health care labor force, and are often responsible for tasks associated with domestic, household labor and skills, such as ancillary and caregiving work; they also often experience unique industrial segregation and occupational polarization. Traditional constructions of paid work and care suggest that men produce while women reproduce, i.e., that men's contributions to the family are defined by their financial contributions while women's contributions to the family are defined by reproduction and care work. However, this study expands upon this literature and uniquely contributes to it because it demonstrates that the financial burden of care for children, households, and transnational families can also be maintained equally, if not more robustly, by women. Nevertheless, it should not be mostly or the sole responsibility of women, but rather it should be a shared obligation; otherwise, it will deplete them of their time, energy, and mental and physical health. In the long run, the quality and sustainability of women's work as well as the social,

economic, and health conditions for these women and their families are at risk if adequate support systems are not put in place.

The findings from this study also pose a contradiction and raise important questions that could be directed for future research. For example, while LTC workers stated that these remittances may be financial support for extended family members, another possibility that could be explored is whether or not remittances are also sent for investment purposes, e.g., acquiring land, buildings, etc. The reason this would be important is because some participants indicated the desire to return to their countries of origin in order to live there. Another area for future research is investigating how workers provide support, given their own costs of living in expensive jurisdictions. One starting point for this type of research would be to ask if workers are taking on loans and debt in order to fulfill these commitments. While LTC workers are entitled to use their income in the way they need/want, the choice to send remittances could also be examined, given that many LTC workers are precariously employed and often underpaid. This also raises questions about the extent to which remittances can be sustained under pandemic and post-pandemic conditions.

**Funding:** A portion of the fieldwork costs for this study received institutional funds for data collection.

**Institutional Review Board Statement:** The study was conducted according to the guidelines of the Declaration of Helsinki, and approved by the Institutional Review Board (Office of Research Ethics) of York University (STU2016-139 and 22 November 2016).

**Informed Consent Statement:** Informed consent was obtained from all subjects involved in the study.

**Data Availability Statement:** The data presented in this study are available on request from the corresponding author. The data are not publicly available due to ongoing research, analysis, and further studies.

**Acknowledgments:** Many thanks to Geoffrey M. Anderson, Rachel Gorman, the editor, and the three anonymous reviewers for their encouragement, guidance, and insightful feedback.

**Conflicts of Interest:** The author declares no conflict of interest. The funders had no role in the design of the study; in the collection, analyses, or interpretation of data; in the writing of the manuscript, or in the decision to publish the results.

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
