# Peer review of "Remittance Flows from Healthcare Workers in Toronto, Canada"

_sustainability, doi:10.3390/su13179536_

Round 1
Reviewer 1 Report
The manuscript is an exploratory study that examines the relations between immigrant workers in the healthcare sector and their remittance behaviours. The current study explores critical dimensions of racialized immigrants and highlights variances in both quantitative and qualitative remittance behaviours of long-term care (LTC) workers. There are some recommendations for the paper. The concerns and issues (in no particular order) are outlined below.
Issues:
|
1. |
Motivation. The paper needs to develop a stronger motivation for the study. Why is the research question relevant or important? The paper motivates the study by exploring: “Do LTC workers send remittances...what motivates the decision to send remittances” (lines 91-94)? These questions are not well motivated theoretically and are too intuitive to be interesting.
|
|
2. |
Motivation. What is the motivation to study healthcare / LTC workers? Should we expect different remittance behaviours between LTC workers and workers from other sectors? What is the theoretical reason for examining LTC workers’ background, gender, visible/non-visible race, and even ancestral background as an implication on remittance behaviour? Perhaps these aspects can be addressed more fervently.
|
|
3. |
Framing. The abstract presents numerous questions related to the implications of remittances and sustainability, incomes and wages, quality of life etc. However, the paper subsequently falls short on delivery. Discussion of the results from the paper in the latter half mainly discussed findings related to value and reasons for remittances. As such, there are substantial gaps regarding your motivation and delivery. What are the implications of your findings, and why are these related to the points mentioned in the abstract and the theoretical framework? For example, you mentioned a gendered society “where women and men have fundamentally different experiences and access to power and privilege” (lines 102-103), but subsequently, your results and discussion do not explore this.
|
|
4. |
Arguments. You used feminist materialist scholarship as the underlying framework and, based on your study, made the argument that “previous research shows that the senders of remittances are often male...originated from female immigrant labor in the care work sector” (lines 433-437). You focused on a sector that is potentially populated by female minorities. There are concerns that the context is the main driver behind the findings and conclusion. Furthermore, the qualitative interviews do not support your statement as the male interviewees and related quotations suggest similar levels of familial support.
|
|
5. |
Arguments. Perhaps you should also consider aspects of reversal causality. These immigrant workers left their home country primarily to support their families. As such, your arguments of remittances as an afterthought related to moral obligation, ethics, desire, and purpose are potentially problematic.
|
|
6. |
Arguments. You considered visible minority as an intersectional concern. This was not discussed. This concern also relates to the broader question – what is the relevance of individual characteristics on remittance outcomes?
|
|
7. |
Methodology. The area related to remittance is rather well-developed. Room for qualitative and exploratory study is limited. Generally, qualitative research should be done for nascent theories and with the purpose of developing new emergent constructs. As such, I think there is a lack of methodological fit due to the state of the present literature and the present findings of the paper.
|
|
8. |
Data. I think it is good practice to publish your survey questionnaire and related interview questions in the appendix. Furthermore, as good tenets of feminist literature, it is vital to practice reflexivity where the objective is presented alongside subjective interpretations.
|
|
9. |
Findings. Related to point 7. The breadth and depth of the paper’s findings is too thin for a relatively matured topic. Discussions related to remittances have been rather well-developed in the literature. An exploratory methodology with findings that suggest variance in both value and reason of remittances leaves more to be desired empirically and conceptually.
|
|
10. |
Implications. There are some gaps related to the findings and conclusion. For example, based on the results, we are not able to ascertain if burdens of care fall disproportionately on women. Information in section 3 (results) does not suggest or support this conclusion.
|
|
11. |
Overall. I believe that there are deeper insights that are not particularly apparent in the present draft. The contribution needs to be clearer. Is the paper’s contribution geared towards feminist theories or remittance literature? Furthermore, the results and findings need more depth to make a substantial contribution.
|
I am thankful and honoured for the opportunity to read your manuscript. I sincerely hope you will find my comments useful for future revisions. I wish you all the best in pursuing this line of work.

Author Response
Hello,
Thank you so much for your feedback and comments which help to strengthen this paper. Please see the attached file.
Kind Regards,

Reviewer 2 Report
If you take the school of thought that the author originates - feminist political economy - from as a reasonable starting point in the competition of differing position on the topic, the article can be published. Nevertheless, one wonders how narrow this starting point is which is hardly challenged by competing approaches that may lead to competing or supplementary explanations.
I rather would take a broader perspective which would check for potential competing explanations that may challenge the adopted approach and may lead to similarly good explanations for the evidence shown in the paper, e.g. from institutional / labour economics / economics of migration or economic sociology / cultural studies or political studies. In other words, the theoretical framework starting from line 96 is quite narrow and does not bother to reject alternative approaches as not applicable / weaker.
The evidence based on n = 43 on the following pages is well structured, nevertheless much is interpreted from such small number of respondents. The break-up into differing regions after line 229 shows that the differing cultural background can hardly be differentiated with e.g. just 3 respondents from Europe or 4 only from East Asia. Discussing then the role of male/female in lines 259 to 262 may have a bias and appears superficial without sufficient backgound. (Particularly if you just look at the LTC jobs with perhaps easier accessibility for e.g. non-native female workers than foreign male immigrants.)
With regard to the evidence in Table 6 in line 234, one also wonders from an economic perspective why nominal $ values (below, above 500$) values are taken despite of very different purchasing power this means for the receiving countries depending on purchasing power in these very different regions (especially because of very different services prices in high real income compared to low income countries etc).
In lines 405 on the answer to the author's question "Why do care workers do this?" does not become fully clear. This is because it could still be the best option from the "transnational families" point of view originating perhaps in the global south, to improve the family's overall living conditions in that way. Then it is not really clear if the reason is "moral responsibility" stemming perhaps from family pressure or if the really feel that their role is "rewarding" since these women may be quite isolated (and may not want to be entirely disconnected from their original/native family backgrounds) and may have to return to their original families in circular migration (line 412), in my opinion.
In lines 504 on one still wonders what would be the alternative for these workers in the countries of origin when they would be nearer to their original families. Perhaps they and their families would be in a still worse situation compared to the current one - at least one should not entirely forget about / neglect the possibility of a win-win solution compared to the most likely alternative situation.
The acknowledgements at the end of the article are somewhat strange since the author thanks already the editor and 3 anonymous reviewers prior to handing in the article to Sustainability.
Author Response

(The authors gave the same response as above.)

Reviewer 3 Report
The recommendations parts of the article should be improved. It should itself be improved in a new version of the manuscript.
Author Response
Hello,
Thank you so much for your feedback. Please see the attached file.
Kind Regards,

Round 2
Reviewer 1 Report
Prior issues of concerns were addressed – either in main text or cover letter. While there remain some concerns with the theoretical framing and positioning of the paper. As a paper with a focused context and an interesting story, I think this is appealing paper for people exploring various forms of inequalities.
Author Response
Hello,
Thank you for your review, it is very much appreciated. Please see the attached.
Kind regards,

Reviewer 3 Report
The manuscript does not need any modification.
Author Response

(The authors gave the same response as above.)
